# The Antibiofilm Effects of Antimony Tin Oxide Nanoparticles against Polymicrobial Biofilms of Uropathogenic *Escherichia coli* and *Staphylococcus aureus*

**DOI:** 10.3390/pharmaceutics15061679

**Published:** 2023-06-08

**Authors:** Inji Park, Afreen Jailani, Jin-Hyung Lee, Bilal Ahmed, Jintae Lee

**Affiliations:** School of Chemical Engineering, Yeungnam University, 280 Daehak-ro, Gyeongsan 38541, Republic of Korea; pij3360@ynu.ac.kr (I.P.); 22140160afreen@yu.ac.kr (A.J.); jinhlee@ynu.ac.kr (J.-H.L.); bilal22000858@yu.ac.kr (B.A.)

**Keywords:** antibiofilm, antimony tin oxide, *Escherichia coli*, polymicrobial biofilms, *Staphylococcus aureus*

## Abstract

Biofilms are responsible for persistent or recurring microbial infections. Polymicrobial biofilms are prevalent in environmental and medical niches. Dual-species biofilms formed by Gram-negative uropathogenic *Escherichia coli* (UPEC) and Gram-positive *Staphylococcus aureus* are commonly found in urinary tract infection sites. Metal oxide nanoparticles (NPs) are widely studied for their antimicrobial and antibiofilm properties. We hypothesized that antimony-doped tin (IV) oxide (ATO) NPs, which contain a combination of antimony (Sb) and tin (Sn) oxides, are good antimicrobial candidates due to their large surface area. Thus, we investigated the antibiofilm and antivirulence properties of ATO NPs against single- and dual-species biofilms formed by UPEC and *S. aureus*. ATO NPs at 1 mg/mL significantly inhibited biofilm formation by UPEC, *S. aureus*, and dual-species biofilms and reduced their main virulence attributes, such as the cell surface hydrophobicity of UPEC and hemolysis of *S. aureus* and dual-species biofilms. Gene expression studies showed ATO NPs downregulated the *hla* gene in *S. aureus*, which is essential for hemolysin production and biofilm formation. Furthermore, toxicity assays with seed germination and *Caenorhabditis elegans* models confirmed the non-toxic nature of ATO NPs. These results suggest that ATO nanoparticles and their composites could be used to control persistent UPEC and *S. aureus* infections.

## 1. Introduction

Biofilms are highly complex structures composed of microbial cells encapsulated in a matrix of extracellular polymeric substances (EPSs). Clinically, biofilms are a major concern because of their inherent tolerance to common antibiotics and host defense systems and are believed to be responsible for 80% of chronic microbial infections [1]. Urinary tract infections (UTIs) are representative of pathologies attributable to biofilm formation, and UTIs are among the most common reasons for administering antibiotics for adults [2]. Furthermore, catheter-associated UTIs account for more than 65% of UTIs [3], and uropathogenic *Escherichia coli* (UPEC) are responsible for ~80% of infections [4]. In addition, studies have shown that other bacteria, such as *Staphylococcus aureus*, are associated with UTIs. *S. aureus* accounts for about 13% of UTIs and predominantly affects elderly individuals residing in hospitals [5]. *S. aureus* adheres to uroepithelial cells and forms biofilms within the urinary bladder [6]. Moreover, dual-species biofilms formed by UPEC and *S. aureus* have been reported to be responsible for catheter-associated UTIs [7,8]. However, despite the well-established link between bacteria and UTIs, few studies have explored the system of dual-species biofilms [9,10].

Nanoparticles are considered to be promising for the treatment of biofilms, largely because they seem to be unaffected by antibiotic resistance mechanisms [11]. A diverse range of metal nanoparticles, such as silver, gold, zinc, copper, gallium, iron, etc., have been shown to possess antibacterial and antibiofilm activities against various microorganisms due to their nanometer-scale sizes and large surface area-to-volume ratios, which can be used to control microbes’ physical and chemical properties [12,13]. Furthermore, it has been reported that metal oxide nanoparticles, such as AgO, AuO, ZnO, CuO, Ga_2_O_3_, Fe_3_O_4_, etc., possess better biological activities than their respective parent metal ions [14]. Hence, other kinds of metal oxide nanoparticles, such as Al_2_O_3_, Co_3_O_4_, SiO_2_, and TiO_2_, and their complex forms have been further investigated to find novel antibiofilm agents. Recently, tin oxide and antimony oxide nanoparticles have also been proved to show anticancer and antimicrobial properties. For example, antimony oxide (Sb_2_O_3_) nanoparticles at high concentrations (15–60 mg/mL) exerted antimicrobial activity against *E. coli* [15], whereas tin oxide (SnO_2_) nanoparticles had MICs of 0.5 and 8 mg/mL against *E. coli* and *Candida albicans*, respectively [16]. Antimony-doped tin oxide nanoparticles (ATO NPs) are extremely stable and used in aerospace and electrochemical applications [17]. Furthermore, ATO nanocrystals were found to kill cancer cells efficiently and to be non-toxic to mice [18]. Most recently, ATO NPs showed photocatalytic and antibacterial activities with MICs ranging from 25 to 100 mg/mL [19]. However, their antibiofilm and antivirulence activities at subinhibitory concentrations have not been studied.

In this study, we investigated the antibiofilm properties of ATO NPs against UPEC*, S. aureus,* and UPEC*/S. aureus* biofilms. Additionally, we tested the effect of ATO NPs on the virulence properties, viz., hydrophobicity, hemolysis, and slime production, of both bacterial species and their ability to adhere to nylon membranes. The biofilm inhibitory effects were visualized with live imaging microscopy and SEM. We also analyzed the gene expression in *S. aureus* after treatment with ATO NPs, and their toxic effects on nematode *C. elegans* model survival and plant *Brassica campestris* growth. This study demonstrated, for the first time, that ATO NPs inhibited biofilm formation of two pathogenic bacteria and their combination and suppressed their virulence characteristics, and partially revealed their action mechanism.

## 2. Materials and Methods

### 2.1. Chemicals and Microbial Culture

ATO NPs (SnO_2_/Sb_2_O_5_ < 50 nm, product code 549541) were obtained from Sigma-Aldrich (St. Louis, MO, USA). The methicillin-sensitive *S. aureus* strain (ATCC 6538) and UPEC O6:H1 strain (CFT073, ATCC 700928) were obtained from the American Type Culture Collection (Manassas, VA, USA). *S. aureus* and UPEC cells were cultured in Luria-Bertani (LB) and nutrient broth (NB) media, respectively, and incubated at 250 rpm. All assays were performed at 37 °C. All microbial assays were performed using at least two independent cultures in triplicate.

### 2.2. Characterization of ATO NPs with SEM, X-ray Diffraction, TEM, DLS, and Zeta-Potential Analyzer

To analyze the size and composition of ATO NPs, ATO NPs were dissolved in 500 µL of ethanol, dropped onto a coverslip (1 mm × 1 mm), dried, platinum-sputtered, and subjected to a scanning electron microscope (SEM) (Hitachi Systems Ltd., Tokyo, Japan) at 15 kV. The composition of ATO NPs was assessed via X-ray diffraction analysis using a PANalytical Empyrean X-ray diffractometer.

Additionally, a transmission electron microscope (TEM), JEOL 100/120 kV TEM (JEOL, Tokyo, Japan) operating at 200 keV voltage was used to further analyze the ATO NPs. Sample preparation involved drying 10 µL of ultrasonicated ATO NPs at 40% amplitude for 20 min on a copper grid at room temperature. This ensured the immobilization of ATO NPs on the grid for observation under TEM. To obtain statistically significant data, the captured micrographs were then examined to determine the percent frequency of size distribution and the average primary particle size.

For dynamic light scattering (DLS) and zeta-potential analysis, a suspension of ATO NPs was prepared by dissolving 40 µg of ATO NPs in deionized distilled water and subjecting it to ultrasonication for 10 min at 40% amplitude. This sonication step ensured proper dispersion of the NPs in the suspension. To assess the hydrodynamic size of the NPs in the suspension, a Zeta Sizer Nano-ZS90 instrument (Malvern, UK) was employed. Furthermore, the ζ-potential values of the NPs were determined to assess their surface charge characteristics. To obtain reliable results, ten readings were taken for each sample, and the presented values represent the average ζ-potential.

### 2.3. Crystal Violet Assay in 96-Well Polystyrene Plates

The susceptibilities of *S. aureus* and UPEC biofilms to ATO NPs were investigated using a crystal violet assay and 96-well polystyrene plates (SPL Life Sciences, Pocheon, Korea), as previously reported [20]. Briefly, to produce single-species biofilms, UPEC or *S. aureus* was diluted with NB or LB media, respectively, at 1:100, whereas to produce dual-species biofilms of UPEC*/S. aureus*, NB, LB, and distilled water were mixed at a ratio 1:1:1 and equal amounts of UPEC and *S. aureus* cultures were added to microtiter plates, as previously reported [21]. ATO NPs were then added at a concentration of 0, 100, 200, 500, or 1000 µg/mL and incubated for 24 h at 37 °C under static conditions. The plates were thoroughly washed with distilled water and stained with 0.1% crystal violet, incubated for 20 min, and washed three times with distilled water. Biofilms stained with crystal violet were dissolved using 95% ethanol, and optical densities at 570 nm using a Multiskan EX microplate reader (Thermo Fisher Scientific, Waltham, MA, USA) were used to quantify the amount of biofilm formation.

To test the biofilm dispersal ability of ATO NPs, biofilms of UPEC or *S. aureus* were established for 24 h, as described above. Additionally, planktonic cells were carefully removed, and fresh medium with ATO NPs was added into the same well and incubated for another 24 or 48 h. The above biofilm assay was performed.

### 2.4. Microscopic Imaging of Biofilms

To observe biofilm formation, live imaging microscopy and SEM were used. Biofilm formation on a polystyrene surface in the 96-well plates were visualized with an iRiS™ Digital Cell Imaging System (Logos BioSystems, Anyang, Republic of Korea). Biofilm formation was obtained as mentioned above and planktonic cells were carefully discarded, and biofilms were stained with 0.1% crystal violet. Then, 2D images and color-coded 3D pictures were reconstructed from images using software ImageJ (version 1.53e, Bethesda, MD, USA). Biofilms grown on nylon membranes were subjected to SEM, as previously described [22]. Briefly, single- and dual-species biofilms of *S. aureus* and UPEC were grown on 3 × 3 mm nylon membranes with and without ATO NP (0, 500, and 1000 µg/mL). Membranes were fixed with 2.5% glutaraldehyde–2% paraformaldehyde at 4 °C overnight, dehydrated using a series of ethanol treatments (30%, 50%, 70%, 90%, and 100%), and kept for 20 h in isoamyl acetate. Critical-point drying was carried out, and the membranes were coated with platinum and visualized using an SEM S-4200 (Hitachi Systems Ltd., Tokyo, Japan) at 15 kV.

### 2.5. Cell Surface Hydrophobicity (CSH)

CSH was analyzed using hexadecane. *S. aureus* and UPEC alone or in combination were exposed to ATO NPs at 0, 100, 200, 500, or 1000 µg/mL for 24 h at 250 rpm shaking at 37 °C. A total of 1 mL of cultures was then centrifuged for 10 min at 10,000 rpm. Cell pellets were resuspended in PBS buffer, and absorbances at 600 nm were recorded (A_0_)_._ Hexadecane (1 mL) was added to PBS suspensions, vigorously vortexed for a minute, and left undisturbed for 30 min at room temperature to separate the aqueous phase from the organic phase. Absorbances of aqueous phases were then measured at 600 nm (A_i_). The following formula was used to calculate percentages of cell hydrophobicity.
% Hydrophobicity = (A_0_ − A_i_)/A_i_ × 100

### 2.6. Sheep Blood Hemolysis Assay

Hemolysis analysis using sheep blood was performed as previously reported [20]. Briefly, *S. aureus*, UPEC, or *S. aureus*/UPEC were treated with ATO NPs (0, 100, 200, 500, and 1000 µg/mL) and cultured at 250 rpm shaking for 24 h at 37 °C. Fresh sheep blood cells (MBcell, Seoul, Republic of Korea) were centrifuged at 3000 rpm for 5 min. Red blood cell pellets were washed five times with PBS buffer, resuspended in 10 mL of PBS buffer with 300 µL of bacterial culture, incubated for 1 h with 250 rpm shaking in 37 °C, and centrifuged at 10,000 rpm for 10 min. Optical densities of the supernatant were measured at 543 nm.

### 2.7. Slime Production by S. aureus

The phenotypic changes in colony formation by *S. aureus* were observed on Congo red agar plate, as previously reported [23]. *S. aureus* culture (1 µL) was plated on Congo red agar containing 36 g/L of sucrose (Sigma), 37 g/L of brain–heart infusion broth (BD Biosciences, Franklin Lakes, NJ, USA), 0.8 g/L of Congo red (Sigma), 15 g/L of agar (BD Biosciences), and 1000 µg/mL of ATO NPs. Additionally, the agar plates were then incubated for 24 h at 37 °C, and changes in colony morphology were visualized.

### 2.8. Gene Expression Study with qRT PCR

To determine whether ATO NPs affect the expressions of biofilm- and virulence-related genes, qRT-PCR was performed. *S. aureus* was grown in 20 mL of LB to an OD_600_ of 1, and then ATO NPs (1000 µg/mL) were added, cultured for 4 h, and RNase inhibitor (RNAlater, Ambion, TX, USA) was added with mixing to prevent RNA degradation. The *S. aureus* cultures were harvested via centrifugation at 13,000 rpm for 10 min, and total RNA was purified from cell pellets using the Qiagen RNeasy mini kit (Valencia, CA, USA). The purity of total RNA extracted was confirmed using a nanodrop spectrophotometer (Fisher Scientific, Loughborough, UK). qRT-PCR was performed using ABI StepOne Real-Time PCR System (Applied Biosystems, Foster City, CA, USA) and SYBR green master mix (Applied Biosystems). Differential expressions of the *agrA*, *aur*, *hla*, *icaA*, *nuc1*, *RNAIII*, *saeR*, *sarA*, *seb*, *sigB*, and *spa* genes were evaluated. *16s rRNA* was used as a housekeeping gene. The primers used are listed in Appendix A. At least two independent cultures with four reactions were used.

### 2.9. C. elegans Toxicity Study

The nematode *C. elegans* model has been widely used to study biomedical and nanotoxicology. The toxicity of ATO NPs in the nematode *C. elegans* was investigated as previously reported [24]. Briefly, the synchronized *C. elegans* strains *fer-15(b26)* and *fem-1(hc17)* were washed with M9 buffer, and approximately 30 worms were added to each well of a 96-well plate containing different concentrations of ATO NPs (0, 100, 200, 500, and 1000 µg/mL). Additionally, the plates were incubated for 8 days at 25 °C without shaking. Numbers of dead and surviving worms were counted daily. Nematode survival was observed using the iRiS^TM^ Digital Cell Imaging System (Logos BioSystems, Anyang, Republic of Korea).

### 2.10. Effect of ATO NPs on Plant Brassica campestris Growth

The toxicity of ATO NPs on plant growth was also investigated. *B. campestris* seeds were rinsed with distilled water three times and left to soak for 5 h, and 10 seeds were plated in Murashige and Skoog plant growth medium containing 0.7% agar [22] and different concentrations of ATO NPs (0, 100, 200, 500, and 1000 µg/mL). Then, the agar plates were incubated for 6 days at room temperature, and plant lengths were measured.

### 2.11. Statistical Analysis

The analysis was conducted using Student’s *t*-test, and graphs were plotted using SigmaPlot 14.0 (Systat Software, Inc., Chicago, IL, USA). Experiments were performed using at least two independent cultures, and results are presented as means ± standard deviations. *p* values of <0.05 were accepted as significant changes and are marked with asterisks.

## 3. Results

### 3.1. Characterization of ATO Nanoparticles

The size and composition of ATO NPs were initially characterized. Images of ATO NPs were captured using SEM, and the average particle size, as determined by ImageJ, was 31 ± 3 nm (Figure 1A,B). X-ray diffraction analysis showed ATO NPs were composed of 57.26% tin (Sn) and 5.32% antimony (Sb) (Figure 1C,D), which agreed with the characteristic description supplied by Sigma Aldrich (Appendix A). The morphology of ATO NPs was also examined using TEM, revealing a uniform shape with individual aggregates and a rough surface (Appendix A). TEM micrographs revealed the average mean particle diameter of ATO NPs to be 16 ± 5 nm (Appendix A). Measurements conducted in distilled water indicated the presence of particle aggregates with a hydrodynamic size of 261 ± 13.6 nm (Appendix A). This increase in size compared to the primary particle diameter suggested some level of aggregation in the distilled water. It is noteworthy that the ATO NPs demonstrated relative stability in the suspension. The zeta potential analysis showed a negative surface charge with a ζ-potential value of −31 ± 1.7 mV.

### 3.2. Inhibition of Single- and Polymicrobial-Biofilm Formation by ATO NPs

The biofilm inhibitory effects of ATO NPs on mono- and dual-species biofilms were examined at 0, 50, 100, 200, 500, and 1000 µg/mL. ATO NPs dose-dependently decreased biofilm formations by UPEC and *S. aureus* (Figure 2A,B). *S. aureus* biofilms are more sensitive to ATO NPs than UPEC biofilms. For example, *S. aureus* biofilm formation was inhibited by 44 and 91% at ATO NP concentrations of 200 and 1000 µg/mL, respectively, and UPEC biofilm formation was inhibited by 42 and 67% at 200 and 1000 µg/mL, respectively.

Furthermore, decent dual biofilms of UPEC and *S. aureus* cells were formed in the mixture of NB, LB, and water (1:1:1). It appears that biofilm formation (~1.8 at OD_570_) of untreated dual cultures (Figure 2C) was in the middle of the single-biofilm formation of *S. aureus* (~3.7 at OD_570_ in Figure 2A) and UPEC (~1.4 at OD_570_ in Figure 2B). Dual-species biofilm formation was inhibited to a lesser extent, specifically by 38 and 60% at ATO NP concentrations of 200 and 1000 µg/mL, respectively (Figure 2C). However, ATO NP concentrations of up to 1000 µg/mL did not inhibit the planktonic cell growth of *S. aureus* or UPEC (Appendix A). This result partially agrees with previous antibacterial results with MICs ranging from 25 to 100 mg/mL [19]. These results show ATO NPs at sub-minimum inhibitory concentrations suppressed biofilm formation by *S. aureus* and UPEC without inhibiting planktonic cell growth. The biofilm IC_50_ of ATO NPs for single biofilms of UPEC or *S. aureus* and dual biofilms was 391 µg/mL, 549 µg/mL, or 707 µg/mL, respectively. However, ATO NP concentrations of up to 2000 µg/mL could not eradicate the preformed biofilm of single or dual biofilms of UPEC and *S. aureus* (Appendix A), which confirmed that biofilm dispersal was much more difficult than biofilm inhibition.

### 3.3. Visualization of Biofilms after ATO NP Treatment

To confirm biofilm reduction, live cell imaging and SEM were used. Three-dimensional plots demonstrated that ATO NPs at 100–1000 µg/mL dose-dependently inhibited *S. aureus* biofilm formation on polystyrene surfaces (Figure 2D). Additionally, SEM images showed only sparse cells of round-type *S. aureus* or rod-type UPEC on membranes at ATO NP concentrations of 500 and 1000 µg/mL (Figure 3). Additionally, the presence of ATO NPs was clearly observed around two microbes. In the non-treated control of dual biofilms, a mixture of *S. aureus* cells and UPEC cells was observed, and the treatment of ATO NPs markedly decreased the number of mixed cells. However, it appears that no morphological changes in *S. aureus* or UPEC were observed with treatment of ATO NP concentrations up to 1000 µg/mL. The results indicate that ATO NPs exerted antibiofilm activity without killing the two microbes.

### 3.4. Effects of ATO NPs on CSH

CSH is an essential virulence factor as it directly influences adhesion to biotic and abiotic surfaces. It appears that *S. aureus* was more hydrophobic than UPEC, and ATO NPs had no significant effect on the CSH of *S. aureus* at concentrations of ≤1000 µg/mL (Figure 4A,B). However, ATO NPs dose-dependently reduced the CSH of UPEC (Figure 4B), and at 1000 µg/mL they significantly reduced the CSH of *S. aureus/*UPEC (Figure 4C). Hence, ATO NPs affected the CSH of UPEC but not of *S. aureus*.

### 3.5. Impact of ATO NPs on Hemolysis

*S. aureus* and UPEC cause hemolysis by producing α-toxin, which is also considered important for virulence and biofilm formation [25]. ATO NPs at 500 and 1000 µg/mL reduced hemolysis by *S. aureus* by 60% and 95%, respectively (Figure 4D), but not that by UPEC. Interestingly, ATO NPs more significantly reduced the hemolytic activity of dual cultures of *S. aureus*/UPEC. For example, 94% and 95% reductions were observed at concentrations of 500 and 1000 µg/mL, respectively (Figure 4E). However, ATO NPs alone at a concentration of up to 2000 μg/mL without bacteria did not cause any hemolysis (Appendix A).

### 3.6. Slime Production by S. aureus

Slime is produced by coagulase-negative *Staphylococcus* and is considered a virulence factor because it increases the ability to colonize [26]. The colony phenotypes produced in the presence of ATO NPs at 1000 µg/mL were strikingly different from those produced in its absence (Figure 4F). The colony color of the untreated sample was dark black due to Congo red staining slime, while the ATO-treated colony appeared to be pale pink, indicating a lower amount of slime production. These observations demonstrated the slime-depressing characteristics of ATO NPs, and ATO NPs did not much affect the colony growth at 1000 µg/mL of ATO NPs.

### 3.7. Gene Expression Analysis

Various genes of *S. aureus* regulate biofilm formation and virulence, and hence, we studied the effect of ATO NPs on gene expressions in *S. aureus* (Figure 5). qRT-PCR showed that ATO NPs at 1000 µg/mL downregulated expression of the *hla* gene by 4.3-fold. This gene is responsible for the production of α-toxin, a cytotoxin that causes hemolysis and has been proposed to play an important role in biofilm formation [25]. The repression of *hla* gene expression supports the inhibition of hemolytic activity by ATO NPs (Figure 4D,E). However, ATO NPs upregulated the expressions of *aur* and *icaA* (responsible for immune evasion and polysaccharide production, respectively) by 3.3- and 2.9-fold. Additionally, ATO NPs had little impact on the expression of other genes.

### 3.8. Toxic Effects of ATO NPs on Plant Growth and Nematode Survival

The toxicological impacts of ATO NPs were investigated with plant and nematode models. ATO NPs at a concentration of ≤1000 µg/mL had no effect on *Brassica campestris* seed germination and growth (Figure 6A,B). In fact, 100% germination and no reduction in seedling length were observed at all tested concentrations. Similarly, ATO NPs at 500 and 1000 µg/mL had only slight impacts on the survival of *C. elegans* (Figure 6C). After 8 days of exposure, ATO NPs at 1000 µg/mL only reduced percentage survival by 19% compared to the non-treated controls. (Figure 6C). These results show ATO NPs are non-toxic to *C. elegans* and plant growth.

## 4. Discussion

Microbial colonization is responsible for various protracted biofilm-associated diseases. Most microbes coexist and produce EPSs, which enables them to form biofilms in environmental niches and microbial diseases [27]. *S. aureus* and UPEC are recognized causative agents of UTIs and can form dual-species biofilms in the urinary tract and on catheters. On the other hand, various metal oxide nanoparticles have potent antimicrobial activities, which has been attributed to their high ratios of surface area-to-volume and abilities to penetrate cells [13,28]. We investigated the ability of ATO NPs to inhibit biofilm formation by Gram-positive *S. aureus*, Gram-negative UPEC, and *S. aureus*/UPEC in combination. Interestingly, ATO NPs did not affect the cell growth of *S. aureus* or UPEC at ≤1000 µg/mL but inhibited biofilm formation by *S. aureus*, UPEC, and *S. aureus*/UPEC (Figure 2). This property is highly beneficial as the antibiofilm activity without affecting planktonic cell growth would result in less development of drug resistance [29].

The antibiofilm activity of ATO NPs on polystyrene and nylon membranes (Figure 1 and Figure 2) may have been caused by interactions between Sb^3+^ and Sn^2+^ ions and negatively charged biofilm EPSs and cell membranes [30], facilitating deeper penetration and localization of nanoparticles. For example, Fulaz et al. reported that positively charged quantum dots penetrated and inhibited *E. coli* biofilms more than neutral or negatively charged quantum dots [31]. Another possible explanation is that biofilms are more susceptible to certain metals, as was demonstrated by Harrison et al., who showed that Sn^2+^ eradicated biofilms formed by *S. aureus* and *E. coli* at concentrations of 17.3 mM and 17 mM, respectively [32]. Furthermore, Sb metal at 15.3 mg/mL showed antimicrobial activity against *S. aureus* [33].

Cell hydrophobicity is a virulence factor that aids the adhesion of bacteria to various surfaces. Studies have shown that *S. aureus* is more hydrophobic than *E. coli* [34], but surprisingly, ATO NPs reduced the CSH of UPEC and *S. aureus*/UPEC but not that of *S. aureus* (Figure 4A–C), possibly because ATO NPs penetrated cells better in a less hydrophobic environment with UPEC [35]. Hence, the change in the CSH of UPEC can be partially attributed to biofilm inhibition. Further study is required to understand the molecular mechanism.

Slime production is important for the adhesion of *S. aureus* to surfaces and for cell-to-cell adhesion and protects bacteria from harsh environments [23], and ATO NP treatment reduced slime production (Figure 4E). Liang et al. showed an AgWPA NP reduced slime production, which was probably due to downregulation of the *icaA* gene [36], but interestingly, this gene was not downregulated by ATO NPs, whereas *hla* was downregulated (Figure 5). Hemolysis is an important virulence factor of *S. aureus* and is regulated by the production of α-toxin, which lyses red blood cells and destroys white blood cells and platelets [37]. We found ATO NPs significantly reduced the hemolytic activity of *S. aureus* (Figure 4D), and this was supported by the ATO NP-induced downregulation of *hla* (Figure 5) and by a study conducted by Saghalli et al., which showed that zinc oxide nanoparticles at sub-minimum inhibitory concentrations decreased hemolysis by reducing *hla* expression [38]. Therefore, the decrease in slime production and hemolytic activity are possible causes of biofilm inhibition in *S. aureus*. Additionally, further investigation on its effects on other bacteria, fungi, and their combination would be of interest.

Currently, nanoparticle toxicity is a major concern, as many studies have reported its negative effects on health and the environment [39]. Antimony (Sb) [40] is a toxic trace element. Additionally, it has been recently reported that its oxide nanoparticles (Sb_2_O_3_·SnO_2_, called ATO NPs here) caused some toxicity in epithelial cell lines [41], while they showed a negligible effect on zebra fish growth [19]. In this study, ATO NPs at concentrations up to 1000 µg/mL did not cause blood hemolysis (Appendix A) and showed no toxicity in plant and nematode models (Figure 6A,C).

In this study, we found that ATO NPs at concentrations up to 1000 µg/mL had no toxic impact on *C. elegans* survival or *Brassica campestris* seed germination (Figure 6). Since most nanomaterials face possible nanotoxicity, a more thorough toxicological study of ATO NPs is required in mammalian models. As various metal nanoparticles have been investigated to treat biofilm-associated infections [13,42], the current study suggests for the first time that ATO NPs have a potential use as an antibiofilm and anti-infectious agent against recalcitrant *S. aureus* and UPEC. Additionally, ATO NPs can be used as medical implant materials and constituents of antibiofilm coating.

## Figures and Tables

**Figure 1 pharmaceutics-15-01679-f001:**
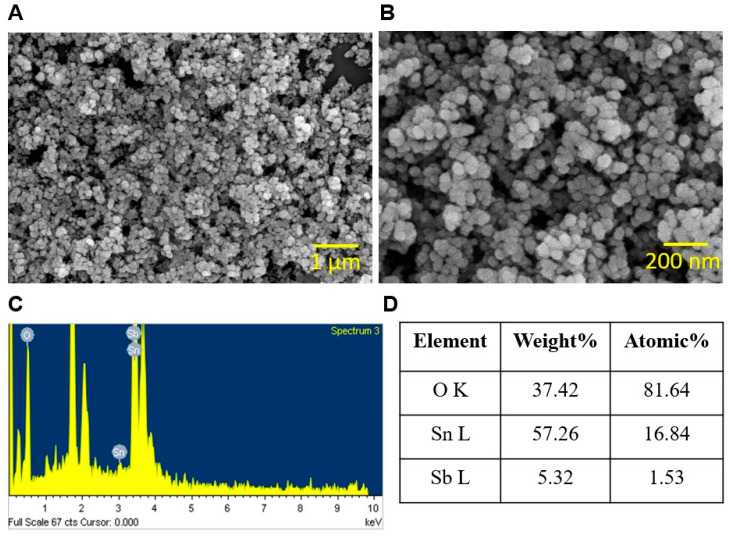
Characterization of ATO NPs using SEM (**A**,**B**). (**C**,**D**) show X-ray diffraction results.

**Figure 2 pharmaceutics-15-01679-f002:**
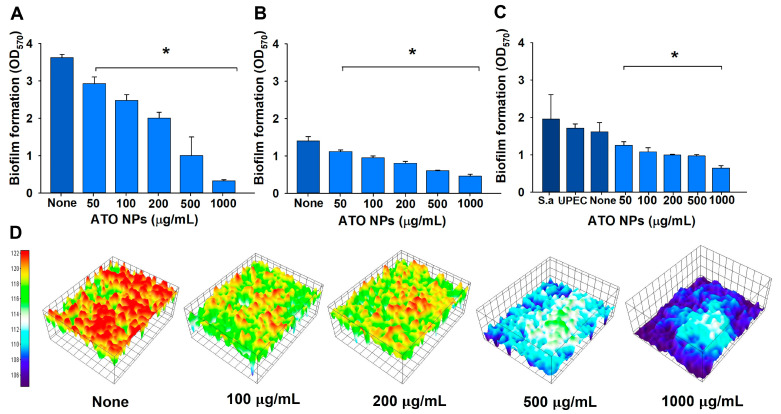
The antibiofilm effect of ATO NPs on *S. aureus* in LB medium (**A**), UPEC in NB medium (**B**), and *S. aureus*/UPEC in the mixed medium (**C**), as determined via the crystal violet assay. S.a indicates *S. aureus*. *S. aureus* biofilm inhibition by ATO NPs on polystyrene surfaces presented as 3D plots (**D**). * *p* < 0.05 vs. non-treated controls (none).

**Figure 3 pharmaceutics-15-01679-f003:**
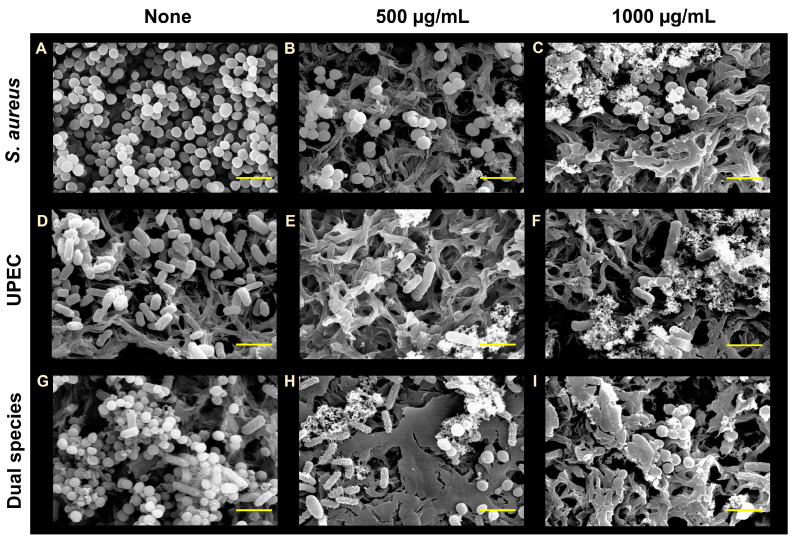
SEM observation of biofilm formation by *S. aureus*, UPEC, and *S. aureus*/UPEC. Untreated (**A**,**D**,**G**) and the presence of ATO NPs at 500 µg/mL (**B**,**E**,**H**) and 1000 µg/mL (**C**,**F**,**I**). The scale bar represents 3 µm.

**Figure 4 pharmaceutics-15-01679-f004:**
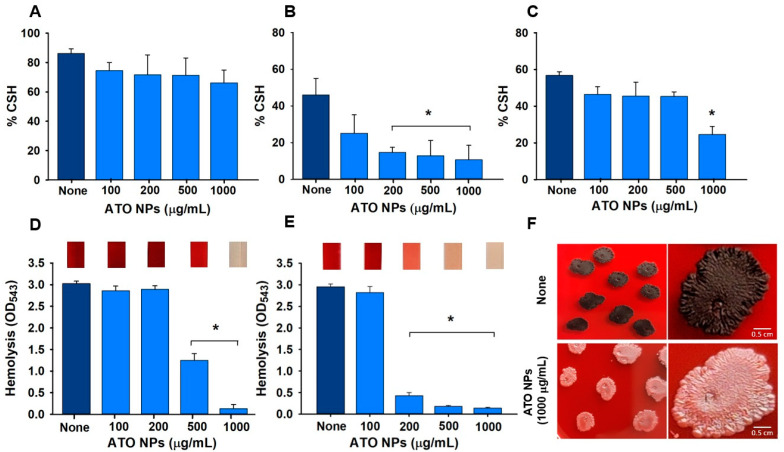
Effects of ATO NPs on CSH of *S. aureus* (**A**), UPEC (**B**), dual species (**C**), hemolysis of *S. aureus* (**D**) and *S. aureus/*UPEC (**E**), and slime production by *S. aureus* on Congo red plates. (**F**). * *p* < 0.05 vs. untreated controls (none). Pictures in (**D**) and (**E**) are supernatant images in cuvettes after hemolysis.

**Figure 5 pharmaceutics-15-01679-f005:**
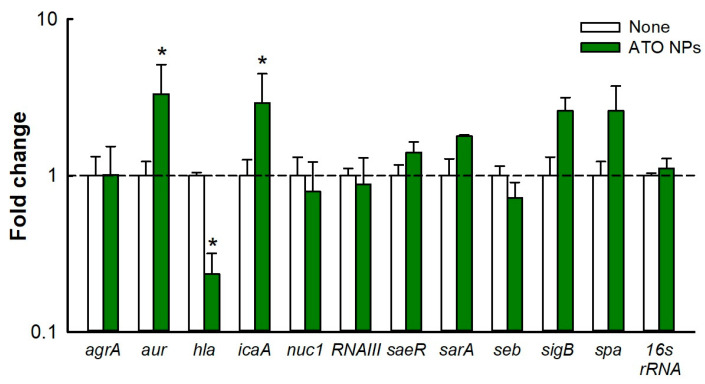
Differential gene expression in *S. aureus* after treatment with ATO NPs (1000 µg/mL). *16s rRNA* was used as the housekeeping gene. Fold changes in gene expression were calculated using the 2^−ΔΔCt^ method compared to the expression of non-treated controls. * *p* < 0.05 vs. non-treated controls (none).

**Figure 6 pharmaceutics-15-01679-f006:**
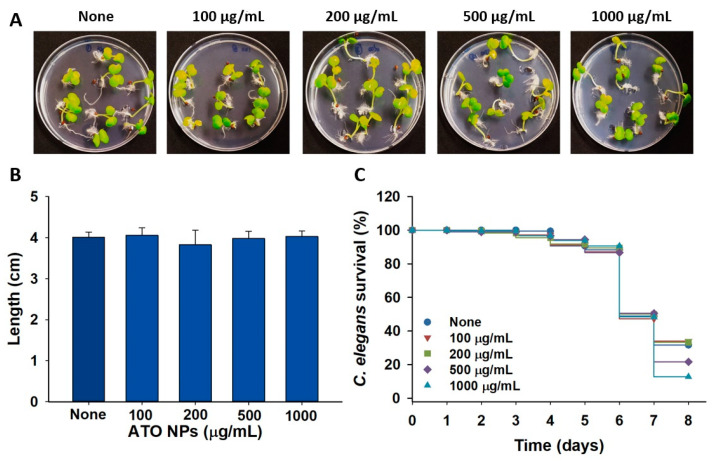
Toxic effects of ATO NPs on *Brassica campestris* seed germination (**A**), seedling length (**B**), and *C. elegans* survival (**C**).

## Data Availability

The data supporting the findings of this study are available within the article and its Appendix A.

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
