# Peer review of "The Antibiofilm Effects of Antimony Tin Oxide Nanoparticles against Polymicrobial Biofilms of Uropathogenic Escherichia coli and Staphylococcus aureus"

_pharmaceutics, 2023, doi:10.3390/pharmaceutics15061679_

Round 1
Reviewer 1 Report
The manuscript “The antibiofilm effects of antimony tin oxide nanoparticles against polymicrobial biofilms of uropathogenic Escherichia coli and Staphylococcus aureus” by Jailani and coworkers evaluates the antibiofilm, anti-virulence, and toxicity of antimony doped tin oxide nanoparticles (ATO NPs). In the current form, it should not be considered for publication since it must be complemented with additional experimental work, it lacks the use of controls (positive and negative) and needs to address the appropriate orthographic corrections. In addition, it is missing conclusions from authors. My comments are presented in the following paragraphs:
In Introduction: please explain why ATO NPs are superior to other types of nanomaterials and why they can exert therapeutic properties such as anticancer, antioxidant, and antimicrobial. On the other hand, C. elegans was used as model to test the toxicity of ATO NPs, please add information that explains the importance of this nematode in the nanotechnological and biomedical pipeline. In L. 63, please explain what is the subinhibitory concentration and which is its range; please add references.
In section 2.2. Characterization of ATO NPs by SEM and X-ray diffraction: in L. 85, change “died” to “dyed”. The acronym of SEM is evidently known, please add the complete name of this technique. Even though ATO NPs were purchased, analyses by dynamic light scattering, UV-Vis spectroscopy, and FTIR spectroscopy can improve information about material characterization.
In section 2.3. Crystal violet biofilm assay: in L. 91, crystal violet assay in a 96 well polystyrene plates. Do the concentrations tested in this study (0-1000 ug/mL) belong to the subinhibitory range? What were the controls used to compare the antibiofilm activity of ATO NPs?
In section 3.2. Inhibition of single- and polymicrobial-biofilm formation by ATO NPs: this section can be expanded if authors explain why the antibiofilm activity of ATO NPs is different against UPEC and S. aureus. If a t-test was used to assess statistical differences between treatments, Figure 2 is illegibly designed since each one of the graphs must be compared against cultures without treatment. Again, non-treated controls are not enough to confirm the antibiofilm activity of ATO NPs. Can authors re-execute this experiment with antibiotics as positive control?
In section 3.3. Visualization of biofilms after ATO NPs treatment: SEM images are interesting as they prove the biofilm inhibition with ATO NPs treatments. However, it is missing the rest of the concentrations mentioned in L. 161; this can be complemented with supplementary material.
In Figure 5 What are the units of Fold change?
The term cell hydrophobicity is called CSH in L.221, please make it uniform throughout manuscript.
Please conclude your findings and present the future directions of this work.
Reviewer 3 Report
- The cytotoxicity assay that you performed on C. elegans it’s not quite correct because you see, you have NP with stibium and other heavy metal which is quite dangerous. Antimony is considered to be one of the most toxic of the heavy metals, and therefore has lower limits than other metals except cadmium (Cd) and mercury (Hg). Yes, it is the conjugated form but still, you should’ve ran that test for a longer period of time to observe if these heavy metals can be excreted correctly and that they don’t accumulate and exacerbate other malignancies.
- For the characterization of the nanoparticles, you should’ve used more techniques that back up your claims of stability (Transmission Electron Microscopy (TEM), Dynamic Light Scattering (DLS), Zeta Potential. Because antimony is stable in space it doesn’t mean that it can be stable in a NP form in the human body conditions.
- Also, when you talk about the effects of ATO NPs on cell hydrophobicity, you claim that S aureus are not that hydrophobic and that you have to increase the concentration of the nanoparticle to 1000 µg/mL, that is a high concentration, too high. Again, I am stressing the problem of antimony accumulation in the organism.
Reviewer 4 Report
In this manuscript, the author developed antimony doped tin (IV) oxide (ATO) NPs to inhibit biofilms formation in E.coli. and Staph. aureus. Particle size was measured by SEM, and composition was evaluated by X-ray diffraction. Then ATO NPs with different concentrations was applied to E. coli and Staph. Aureus to evaluate biofilm inhibition, their cell hydrophobicity effects, hemolysis, slime production and gene expression change. Finally, nonsignificant toxicity of ATO NPs was evaluated on C.elegans. The experimental design is clearly to demonstrate that ATO NPs can inhibit biofilm growth of E. coli and Staph. Aureus, however, there are some concerns need to be addressed.
1. Please correct ‘died’ on line 85 to dried.
2. Please calculate IC50 of ATO NPs for antibiofilm effect based on Figure 2A-C and compare antibiofilm effect of ATO NPs in S. aureus, UPEC and mixture.
3. Please correct ‘tow’ on line 212 to two.
4. It seems like there is no biofilm formation in image Figure 3A, which is untreated S. aureus. Please confirm the images.
5. The hemolysis results indicate hemolytic toxicity of S. aureus and mixture was reduced after treated with 200-1000 ug/mL ATO NPs. However, there is lack of only ATO NPs control with different concentrations in hemolysis studies. Please add the results of only ATO NPs with different concentrations as a control.
6. Please correct ‘plausible’ on line 290 to possible.
Moderate editing of English language
Round 2
Reviewer 2 Report
The authors have tested the antibiofilm properties of antimony tin oxide nanoparticles against single and dual species biofilms of Staphylococcus aureus and uropathogenic Escherichia coli. SEM revealed the particles ranged approximately 50-100 nm in diameter while crystal violet assays confirmed the nanoparticles ability to inhibit single and dual species biofilms. Newly collected data by the group showed that NPs were not very efffective against established biofilms but the ability to prevent formation is still a good start. ATO NPs displayed an effect on UPEC hydrophobicity along with effects on S.aureus ability to cause hemolysis and produce slime. qRT-PCR studies on S.aureus revealed ATO NPs had an effect on expression of specfici virulence factors. Studies on seed germination and C.elegen toxicity demonstrated the minimal eukaryotic toxicity of ATO NPs. Additional data collected has now shown no damage to RBCs from NPs, which is good.
Please check over for spelling and grammatical errors.
Reviewer 3 Report
Everything that i addressed was taken care of so I agree with present correcte manuscript
Reviewer 4 Report
The revised version addressed all my concerns.